# Correlation between Health and eHealth Literacy and a Healthy Lifestyle: A Cross-Sectional Study of Spanish Primary Healthcare Patients

**DOI:** 10.3390/healthcare11222980

**Published:** 2023-11-18

**Authors:** David García-García, María Julia Ajejas Bazán, Francisco Javier Pérez-Rivas

**Affiliations:** 1Nursing Primary Health Care Service of Madrid, 28004 Madrid, Spain; 2Grupo de Investigación UCM “Salud Pública-Estilos de Vida, Metodología Enfermera y Cuidados en el Entorno Comunitario”, Departamento de Enfermería, Facultad de Enfermería, Fisioterapia y Podología, Universidad Complutense de Madrid, 28040 Madrid, Spain; frjperez@ucm.es (M.J.A.B.); majejas@ucm.es (F.J.P.-R.); 3Academia Central de la Defensa, Escuela Militar de Sanidad, Ministerio de Defensa, 28040 Madrid, Spain; 4Red de Investigación en Cronicidad, Atención Primaria y Promoción de la Salud—RICAPPS—(RICORS), Instituto de Investigación Sanitaria Hospital 12 de Octubre (Imas12), 28041 Madrid, Spain

**Keywords:** health literacy, eHealth literacy, healthy lifestyle, primary healthcare, public health, global health, nursing

## Abstract

Background: Health literacy and eHealth literacy play a crucial role in improving a community’s ability to take care of themselves, ultimately leading to a reduction in disparities in health. Embracing a healthy way of living is vital in lessening the impact of illnesses and extending one’s lifespan. This research delves into the link between the health and eHealth literacy levels of individuals accessing primary healthcare services and investigates how this relates to adopting a health-conscious lifestyle. Methods: The approach involves a cross-sectional examination carried out at a healthcare facility in the Madrid region of Spain, focusing on adult patients who are in need of primary care nursing services. Health and eHealth literacy and a healthy lifestyle were measured using the Health Literacy Questionnaire (HLQ), the eHealth Literacy Questionnaire (eHLQ), and the “PA100” questionnaire, respectively. Results: Only some of the dimensions of the HLQ and eHLQ were significantly related to a healthy lifestyle, predominantly with a very low or low relationship. Dimension three of the HLQ and dimension five of the eHLQ acquired more importance and were positioned as positive predictors of a healthy lifestyle. Conclusions: This study helps comprehend the relationship between health and eHealth literacy and a healthy lifestyle, which provides information that contributes to understanding the factors that might have a higher impact on lifestyles.

## 1. Introduction

A person’s lifestyle can be defined as the set of identifiable behavior patterns determined by the interaction between individual characteristics, social interactions, and socioeconomic and environmental living conditions. A lifestyle is considered healthy when a series of habits are acquired that allow a state of complete physical, mental, and social well-being throughout life [1]. Currently, there are various validated questionnaires for measuring healthy lifestyles. The main ones are the “Healthy Lifestyle Assessment Toolkit”, which analyzes 8 different dimensions of health and lifestyle while providing health recommendations for the user [2]; the questionnaire “Estilo de Vida Saludable (EVS)”, which evaluates 9 dimensions related to healthy lifestyle habits [3]; the “MEDLIFE”, which evaluates 28 items related to the Mediterranean lifestyle and its protective habits [4]; the “Fantástico”, which analyzes 10 dimensions related to a healthy lifestyle through 30 items [5]; and the “PA100” questionnaire, a questionnaire of recent validation in the Spanish population that has been specifically designed to be used in primary care patients, which consists of 5 dimensions that measure a healthy lifestyle from a biopsychosocial perspective. A healthy lifestyle is essential both to prevent and avoid the progression of chronic diseases, their disabling nature, and morbidity and mortality. There are many studies that show that leading a healthy lifestyle throughout life plays a vitally important role in our health, increasing healthy life expectancy without the appearance of severe and disabling chronic diseases and increasing life expectancy and quality in people with multimorbidity [6,7,8]. Additionally, the literature positions health literacy as one of the major conditions for leading a healthy lifestyle [9].

Health literacy is a term introduced in 1970 that can be defined as the personal, cognitive, and social skills that determine the motivation and ability of individuals to gain access to, understand, and use information to promote and maintain good health [10]. The most widely used questionnaires to measure health literacy are the “European Health Literacy Survey Questionnaire (HLS-EU-Q)” and the “Health Literacy Questionnaire (HLQ)”. The former consists of 12 dimensions that evaluate the capacity to access, comprehend, evaluate, and apply information related to health promotion and prevention [11], and the latter offers, through its 9 scales, valuable insights into the competencies and deficiencies of individuals and communities regarding their knowledge and abilities in matters related to health [12,13].

eHealth literacy was initially introduced in 2006 and is defined as the capacity to effectively search for, locate, comprehend, and assess health-related information using electronic resources, subsequently applying the acquired knowledge to address and resolve health-related issues [14]. The primary assessment tools for this concept include the “eHealth Literacy Scale (eHEALS)” and the “eHealth Literacy Questionnaire (eHLQ)”. The former comprises eight dimensions that capture insights into knowledge, usage safety, and perceived proficiencies in eHealth [15]. The latter, on the other hand, encompasses 35 scales intended to offer a comprehensive understanding and evaluation of individuals’ engagement with digital health services from various dimensions [16].

The HLQ and eHLQ were used in this study as they provide insight into individual, social, and cultural determinants among groups experiencing disadvantages. Unlike the HLS-EU-Q and eHEALS, which provide a total score, the HLQ and eHLQ utilize individual scores for each of their dimensions. This allows a more precise analysis of the health determinants that influence them and a better understanding of the population’s distribution of health and eHealth literacy strengths and weaknesses.

Health and eHealth literacy have acquired great importance as our society develops, and their impact on health has been recognized, being both considered to be health determinants. The relationship between health and eHealth literacy is still uncertain, and there are very few studies that analyze this. Some of them conclude that there is no existing correlation [17], while others affirm the contrary [18,19]. In addition to this, current research has only been conducted in very specific and younger populations, and only questionnaires that provide a total score rather than analyzing its different dimensions have been utilized.

Higher levels of health and eHealth literacy improve health outcomes, diminish healthcare costs, increase motivation to seek health information, lessen hospitalization rates, increase pharmacological adherence, reduce health inequities, empower people to be self-sufficient and make informed decisions, and promote the adoption of a healthy lifestyle [14,17,20,21].

These data suggest that perhaps there is an existing relationship between health and eHealth literacy and the adoption of a healthy lifestyle; however, some studies found that improving health literacy increases health-promoting behavior [22], while others found some dimensions to be inversely associated [23]. With regard to eHealth literacy, some studies found a positive correlation when treating it as a total score [18,24], while others only found this in relation to exercise [15]. In fact, some systematic reviews that analyzed these variables concluded that there is insufficient to low and inconsistent evidence in the matter [25,26].

The aim of this study was to analyze the correlation between health and eHealth literacy and a healthy lifestyle in primary care patients of a healthcare center in Madrid (Spain), as well as to examine which sociodemographic and health and eHealth variables influence a healthy lifestyle.

The initial hypothesis of this study is that people with higher health and eHealth literacy levels will have a healthier lifestyle.

## 2. Materials and Methods

### 2.1. Study Design and Study Subjects

A cross-sectional study was designed in accordance with STROBE guidelines [27]. Data were collected from patients receiving nursing care in the primary healthcare sector in a center situated in the Madrid region (Spain); this center was chosen due to data accessibility. The patients normally treated at the center are elderly people with chronic diseases that require control and follow-up (mainly hypertension and diabetes), who need wound healing treatment, the administration of chronic intramuscular injections, or the modification of unhealthy habits (i.e., smoking cessation, sedentary lifestyle, and obesity). Inclusion criteria were being older than 18 years old and signing the informed consent. Exclusion criteria were not speaking the native language, having dementia, having an uncontrolled mental illness, or requiring urgent assistance due to an acute illness. Following the guidelines of a previous study [28], 166 patients were considered to be necessary. Three individuals per day were recruited via systematic probabilistic sampling, which corresponded to the first, fourth, and seventh patients on the list scheduled for care. The sample was split into two age groups (81 individuals in the <65 years old group and 85 individuals in the ≥65 years group) using the P50 (median) as a cut-off point. This was performed so that both groups were as evenly distributed as possible. This distribution also provided the most statistically significant differences. The median age for the <65 years group was 52, while for the >65 years group, it was 78.

### 2.2. Procedures

All participants were provided with an information sheet designed to provide details about the nature of this study before voluntarily signing the informed consent. After its signature, participants were provided with a sociodemographic data collection sheet, the HLQ, the eHLQ, and the PA100 questionnaire in this order. Once collected, these data were transferred onto an Excel file.

Depending on the autonomy of the participants to complete the questionnaires, these were either self-administered or investigator-administered. The proper completion of the questionnaires was checked before the study visit was finished.

### 2.3. Outcome Measures

The “PA100” questionnaire was used to measure the variable healthy lifestyle (measured as a quantitative, discrete variable). The validation process of this questionnaire has been specifically designed for the assessment of healthy lifestyles in the adult population of Spain in the context of primary care. In addition, as it is a very recently developed questionnaire, it guarantees the collection of data that are currently considered most relevant for assessing lifestyles. It consists of 33 items and evaluates 5 dimensions: diet (25 points); physical activity and sedentary lifestyle (20 points); alcohol, tobacco, and consumption of other drugs (25 points); emotional well-being (20 points); and safety and unintentional injuries (10 points). The maximum total score is 100 points. A higher score translates into a better lifestyle, and a lower score a less healthy lifestyle [1].

The HLQ was employed to assess health literacy, treated as a measurable discrete variable. This robust questionnaire was developed using a strategy that prioritized validity [29]. Across different populations [13,27,28,29] and languages [30,31,32,33,34,35,36,37,38,39], it has consistently demonstrated strong reliability and psychometric properties for all its scales, with reported values ranging from 0.77 to 0.90. In the context of this study, the Spanish version of the HLQ was utilized, having undergone testing with patients under oral anticoagulation treatment [23] and those with cardiovascular conditions [40] within primary care settings. Distinguishing itself from other tools, this questionnaire comprehensively gathers information on health literacy, offering a holistic comprehension of its concept. Furthermore, it operates with a constructivist objective: the amassed data can assist in identifying necessary actions for enhancing health literacy and the quality of healthcare and social services. With a total of 9 dimensions and 44 items, the questionnaire is divided into two sections. The first part comprises 5 dimensions assessed through 23 items, which respondents score as strongly disagree, disagree, agree, or strongly agree (numerically translated to scores of 1–4, respectively). The second part encompasses 4 dimensions evaluated through 21 items, scored as cannot do or always difficult, usually difficult, sometimes difficult, usually easy, or always easy (numerically converted to scores of 1–5, respectively). Higher scores indicate strengths, while lower scores indicate greater health needs [13].

The measurement of eHealth literacy was conducted using the eHLQ, treating it as a quantitative discrete variable. This questionnaire has proven good reliability and validity due to its validity-driven methodology [16,41,42]. The Spanish version of the eHLQ was used. Unlike other tools, it reflects the whole definition of eHealth literacy and helps to identify the learning needs in regard to this matter as well as the quality of the eHealth services offered. The questionnaire has a total of 7 dimensions evaluated through 35 items that are scored as strongly disagree, disagree, agree, or highly agree (corresponding to scores of 1–4, respectively). The highest scores reflected strengths and the lowest health needs [16].

Data pertaining to sociodemographic variables were gathered, encompassing categories such as gender (male or female); age (later categorized as ≤65 or >65 years); educational level (ranging from illiterate or incomplete primary education to primary education, secondary education, high school and professional training, or university studies); country of birth (Spain or third country); marital status (single, married, separated, or widowed); occupation (including employee, self-employed, unemployed, retired or pensioner, engaged in unpaid domestic work, or student); and self-assessed health status (ranging from very bad to bad, fair, good, or very good).

### 2.4. Statistical Analysis

The score of each dimension of the “PA100” was obtained from the sum of the score of each item, and the total score was obtained from the sum of the score of each dimension. Following the recommendations regarding the HLQ and eHLQ, the total scores for each dimension were obtained with the mean of the scores of its items, each having the same weighting. These questionnaires do not provide a total score [13,16].

The Kolmogorov–Smirnov test confirmed a normal distribution for health literacy and eHealth literacy and a non-normal distribution for healthy lifestyles. For this reason, the Spearman´s correlation test was used. A multiple correlation analysis was performed between the total score of the “PA100” questionnaire with the HLQ and eHLQ. Another multiple correlation analysis was conducted between the dimension scores of the “PA100” questionnaire and the dimensions of the HLQ and eHLQ. The following scale for correlation coefficients was used: very low (0 < r ≤ 0.19), low (0.2 < r ≤ 0.39), moderate (0.4 < r ≤ 0.59), high (0.6 < r ≤ 0.79), and very high (0.8 < r ≤ 1.0). A multiple regression model was performed with the total score of the “PA100”, the sociodemographic variables, and the HLQ and eHLQ, with the first questionnaire being considered as the dependent variable to analyze the impact that the independent variables have on a healthy lifestyle. This statistical analysis was conducted after creating the dummy variables of the subcategories of the sociodemographic variables. Only the subcategories that had a relevant impact on having a healthy lifestyle were included. All analyses were conducted with the overall participants and stratified by age groups (<65 years and ≥65 years). Significance was set at *p* < 0.05. All calculations were performed using the IBM SPSS Statistics 27TM statistical package.

### 2.5. Ethics Approval

This study was approved by the ethics review boards of the Madrid Primary Care Center Assistance Management Research Commission (protocol code 01/22-c, approved on 24 January 2022) and by the Ethics Committee of the Complutense University of Madrid (protocol code CE_20220120-10_SAL, approved on 20 January 2022).

## 3. Results

### 3.1. Description of the Sample

The total number of participants in this study was 166, of which 45.2% were men and 54.8% were women. The median age was 65 years, with a range of 73 (between 21 and 94 years) and an interquartile range of 26 (between 52 and 78). In regard to education, 8.4% were illiterate or had incomplete primary studies, 31.3% had primary education, 9.6% had secondary education, 25.3% completed high school or had professional training, and 25.3% had a university education. Of those surveyed, 91% were Spanish, with the remaining 9% being of foreign origin. In relation to marital status, 22.3% were single, 53% were married, 6.6% were separated, and 18.1% were widowed. In terms of employment status, 32.5% were employees, 4.2% were self-employed, 2.4% were unemployed, 50.6% were retired or pensioners, 7.2% engaged in unpaid domestic work, 1.8% were students, and 1.2% were not classifiable. Finally, the perceived state of health was very bad in 0.6% of the cases, bad in 3%, regular in 35.5%, good in 53.6%, and very good in 7.2%.

### 3.2. Correlation between Healthy Lifestyle and Health Literacy

The correlation between health literacy and a healthy lifestyle was significant for dimension two, “having sufficient information to manage my own health” (low in overall and <65 years group); dimension three, “actively managing my health” (moderate in overall and <65 years group and low in ≥65 years group), dimension four, “social support for health” (very low in overall and low in <65 years group); dimension five, “appraisal of health information” (low in <65 years group); dimension six, “ability to actively engage with healthcare providers” (low in overall and ≥65 years group); and dimension nine, “understands health information enough to know what to do” (low in <65 years group) (Table 1).

When the variable of a healthy lifestyle was analyzed by its dimensions, a healthy diet was significantly correlated with dimension two, “having sufficient information to manage my own health” (low in <65 years group); dimension three, “actively managing my health” (very low in overall and low in <65 years group); dimension four, “social support for health” (low in <65 years group); dimension five, “appraisal of health information” (low in the overall group and moderate in <65 years group); dimension eight, “ability to find good health information” (low in <65 years group); and dimension nine, “understands health information enough to know what to do” (low in <65 years group). Physical activity was correlated with dimension two, “having sufficient information to manage my own health” (low in overall and <65 years group); dimension three, “actively managing my health” (moderate in overall and <65 years group and low in ≥65 years group); dimension four, “social support for health” (very low in the overall group); dimension five, “appraisal of health information” (very low in overall and low in <65 years group); dimension six, “ability to actively engage with healthcare providers” (very low in overall and low in ≥65 years group); and dimension nine, “understands health information enough to know what to do” (very low in overall and low in <65 years group). Alcohol, tobacco, and consumption of other drugs were correlated with dimension two, “having sufficient information to manage my own health” (very low in the overall group), and dimension three, “actively managing my health” (very low in the overall group). Emotional well-being was correlated with dimension one, “feeling understood and supported by healthcare providers” (low in <65 years group); dimension two, “having sufficient information to manage my own health” (very low in overall and low in <65 years group); dimension three, “actively managing my health” (low in overall and <65 years group); dimension four, “social support for health” (low in overall and <65 years group); dimension six, “ability to actively engage with healthcare providers” (low in all groups); dimension seven, “navigating the healthcare system” (low in overall and <65 years group); and dimension nine, “understands health information enough to know what to do” (low in overall and <65 years group). Safety and unintentional injuries were only correlated with dimension nine, “understands health information enough to know what to do”, with an inverse and very low correlation in the overall group (Table 1).

### 3.3. Correlation between Healthy Lifestyle and eHealth Literacy

The correlation between eHealth literacy and a healthy lifestyle was significant for dimension two, “understanding of health concepts and language” (low in all groups), and dimension five, “motivated to engage with digital services” (low in ≥65 years group) (Table 2).

When the variable healthy lifestyle was analyzed by its dimensions, a healthy diet was significantly correlated with dimension one, “using technology to process health information” (very low in overall and low in <65 years and ≥65 years group), and dimension two, “understanding of health concepts and language” (very low in overall and moderate in <65 years group). Physical activity was correlated with dimension one, “using technology to process health information” (low in <65 years group); dimension two, “understanding of health concepts and language” (low in all groups); and dimension six, “access to digital services that work” (very low in overall and low in <65 years group). Alcohol, tobacco, and consumption of other drugs were inversely correlated with dimension one, “using technology to process health information” (low in the overall group); dimension three, “ability to actively engage with digital services” (very low in the overall group); and dimension six, “access to digital services that work” (low in <65 years group). Emotional well-being was positively correlated with dimension two, “understanding of health concepts and language” (very low in overall and low in <65 and ≥65 years group); dimension six, “access to digital services that work” (low in <65 years group); and dimension seven, “digital services that suit individual needs” (low in <65 years group). Safety and unintentional injuries were not correlated with any dimension (Table 2).

### 3.4. Multiple Regression Model of Healthy Lifestyle

The multiple regression analysis found the positive predictors of a healthy lifestyle to be dimension three of the HLQ, “actively managing my health” in the global and <65 years model, and dimension five of the eHLQ, “motivated to engage with digital services” in the ≥65 years group.

As negative predictors, two dimensions of the eHLQ were identified: dimension three, “ability to actively engage with digital services” in the global model, and dimension six, “access to digital services” in the <65 years model (Table 3).

## 4. Discussion

The correlation analysis between health literacy and a healthy lifestyle, considering its total score, found no relationship with dimension one, “feeling understood and supported by healthcare providers”, and dimension seven, “navigating the healthcare system”, which might indicate that establishing a relationship with a healthcare provider and being able to navigate the healthcare system does not significantly improve someone’s lifestyle. The same happened with dimension eight, “ability to find good health information”, which reflects that contrary to what one might think, being able to find trustworthy and relevant information does not favor leading a healthy lifestyle. The dimension with higher correlation was dimension three, “actively managing my health”, a relationship that was maintained in all analyses but more elevated in the younger group (r = 0.516, *p* = 0.000). This explains why this dimension was included as a predictor of lifestyle in both the global and younger people regression models. A notable finding was that in the older group, this relationship was much lower (r = 0.304, *p* = 0.005) and was not positioned as a positive predictor in the regression analysis. Hence, recognizing the importance of health and being able to take responsibility for its care is of essential relevance to the younger population. This makes sense, as the youth stage is when we develop new habits that will be maintained throughout life [43].

A relationship was also found with dimension two, “having sufficient information to manage my own health”; dimension four, “social support for health”; dimension five, “appraisal of health information”; dimension six, “ability to actively engage with healthcare providers”; and dimension nine, “understands health information enough to know what to do”. It was interesting to note that all those dimensions had a higher relationship in the younger population, with the exception of dimension six, which found no correlation in the young population and a low correlation in the older one. This might support the relevance of a younger age in acquiring health-related knowledge and having the skills and motivation to apply this to their lifestyle, while on the other hand, in the older population, healthcare providers should put their efforts into establishing a relationship in which they feel proactive and empowered and in control.

Contrary to our hypothesis, it is important to highlight that with the exemption of dimension three, “actively managing my health”, all HLQ dimensions had a low or very low correlation with a healthy lifestyle, which may indicate that there are other and more relevant factors that influence our lifestyle. In the regression analysis, being male was positioned as the most prevalent negative predictor, which perhaps could be attributed to the fact that females generally have more private and public body consciousness, which influences their daily habits [44].

According to the literature, being married has a positive influence on a healthy lifestyle, and being separated has a negative one. Being married is believed to be an important factor in modifying lifestyles, along with having children. This is probably due to the fact that, with this commitment, there is a modification in the beliefs and concern about someone’s own health, as they are more self-conscious about healthy aging and serving as an example for the rest of their family members [45].

Being retired or pensioner has a positive impact on a healthy lifestyle, which may be due to these populations having more free time to take care of their health as well as feeling the need to do so to prevent chronic illnesses related to old age. This is supported by the literature, which mentions that time is the main barrier for younger populations to acquire healthier habits [44].

People who were born in a third country had a less healthy lifestyle, which may be the result of these individuals being more vulnerable as since most of their social support is far away, they might have fewer resources.

The correlation analysis between health literacy and a healthy lifestyle, considering its dimensions, found emotional well-being to be the dimension that most related with the HLQ, followed by a healthy diet and physical activity equally, alcohol, tobacco, and consumption of other drugs, and safety and unintentional injuries in descending order, which highlights the importance that mental health poses on health literacy and vice versa [46].

It was interesting to notice that a healthy diet was related to dimension two, “having sufficient information to manage my own health”; dimension three, “actively managing my health”; dimension four, “social support for health”; dimension five, “appraisal of health information”; dimension eight, “ability to find good health information”; and dimension nine, “understands health information enough to know what to do” in younger population, while none of the HLQ dimensions had any correlation in the older population. This perhaps means that older people’s eating habits are independent of their health literacy levels and are related to other sociodemographic and cultural factors.

In this study, being male was the only sociodemographic variable found to have an impact, as this population has a less healthy lifestyle. This might be due to the female population being most conscious about their health and the male older generations being more prone to smoking and drinking alcohol [47].

Something similar happens with the dimensions of physical activity and emotional well-being, in which the younger population has more HLQ dimensions correlated with them having these in addition to a stronger relationship.

In relation to alcohol, tobacco, and consumption of other drugs, only dimension two, “having sufficient information to manage my own health”, and dimension three, “actively managing my health”, were correlated with a very low correlation, which increases the importance of these factors in preventing its consumption.

Safety and unintentional injuries were only correlated with dimension nine, “understands health information enough to know what to do”, which was inversely correlated. These might imply that people with more education or health education are prone to take more risks or perceive the risk as not as bad. In this case, it is possible that health literacy and its dimensions have acquired less importance since older people have a tendency to relate passively to health information and the health environment, letting themselves be guided by the opinions of their family and/or health professionals. This could indicate that their lifestyle becomes related to the level of health literacy of their primary caregivers.

The results associated with health literacy in the group of older people do not align with the results of the majority of studies that analyze this, such as Cabellos-García et al. [48] obtaining a significant relationship between health literacy and management of anticoagulant treatment and frequency of complications. Similar results have been found in other studies on health literacy related to knowledge and management of oral anticoagulation [49,50] and another that associated health literacy with diabetes control [51]. This may be due to the fact that the level of literacy was analyzed in a group of people (patients with a chronic disease diagnosed for years of evolution) and related to very specific situations (adequate management of oral anticoagulation, knowledge about oral anticoagulation) instead of a concept as general as a healthy lifestyle. In fact, Neter and Brainin [26], in their systematic review, highlighted the insufficient evidence that exists regarding the relationship of health literacy with the perception of one’s own health and health outcomes, in addition to finding a non-association between health literacy and quality of life.

The absence of studies studying the relationship between lifestyle and health literacy could indicate that health literacy acquires importance in specific patients and diseases for more concrete purposes and interventions such as disease control but not being as useful for the general population or concepts as broad as healthy lifestyle. It is possible that this relationship found in the literature, mainly in chronic patients who have to keep good control of their disease, suggests that their good habits are not due so much to their health literacy but rather to the high-risk perception they have knowing that poor control of your disease could worsen your quantity and quality of life.

On the contrary, it was unexpected to find in the rest of the age groups that, although an association between health literacy and a healthy lifestyle was demonstrated, all the correlations had such a low association, except for dimension three (actively managing my health). It can indicate that there are other factors not seen in this study that influence a healthy lifestyle much more. In fact, an experimental study conducted a health education program on healthy lifestyles [52], and although the program did improve health literacy levels, it did not lead to significant changes in healthy lifestyles. In addition, these results reinforce the aforementioned theory that states that a healthy lifestyle may be mainly due to the motivation to manage and take care of one’s own health, the social context, the purpose of life, beliefs, and attitude towards health, the assessment of one’s own health and the perception of risk; that is, health behaviors. In this way, traditional beliefs related to stress or lack of time as the main barriers may acquire much less importance. This conclusion is already supported by the literature [53] and some studies [54,55]. That is why social determinants acquire great importance in lifestyle, which is also in accordance with well-known theories such as Dorothea Orem [56] and self-care, stating that to carry out a certain behavior, it is not enough to have the knowledge of it, but it is necessary to have the attitudes, aptitudes, and motivation necessary to achieve it.

The correlation analysis between eHealth literacy and a healthy lifestyle, considering its total score, only found a significant relationship in dimension two, “understanding of health concepts and language”, and dimension five, “motivated to engage with digital services”. Dimension two, being significant in all groups but more elevated in the younger group (r = 0.309, *p* = 0.005), might indicate that, as well as health literacy, some concepts of eHealth literacy could be crucial to be taught at a younger age. Additionally, dimension five, being significant only in the older group (r = 0.287, *p* = 0.008), could be attributed to the fact that since older people are not digital natives, their motivation to engage with them is the factor that most influences them. These data are supported by the multiple regression analysis that positioned this dimension as the only predictor in the older people group, having a positive impact. However, it was unexpected to observe that the rest of the eHLQ dimensions did not have a correlation with a healthy lifestyle, although dimension three, “ability to actively engage with digital services”, and dimension six, “access to digital services”, were positioned as positive predictive factors in the younger group. These data support the idea that although the results show that there is little or no relationship between these two variables (contrary to our hypothesis), some of its areas might be relevant to be considered in the young population.

The correlation analysis between eHealth literacy and a healthy lifestyle considering its dimensions found that all “PA100” dimensions were similarly related to the eHLQ with the exception of dimension five, “safety and unintentional injuries”, which makes sense as these dimensions analyze physical injuries that might result when no taking caution.

A healthy diet and physical activity were correlated with dimension one, “using technology to process health information”, and dimension two, “understanding of health concepts and language”, having a stronger relationship in the younger population. Although the correlations are low to moderate, it highlights the importance of these concepts to be developed at an early age to be able to manage health appropriately.

Similarly, emotional well-being was correlated with dimension two, in addition to dimension six, “access to digital services”, and dimension seven, “digital services that suit individual needs”, in the younger group, which perhaps highlights the importance of the appropriate use of technologies in these population so as it does not have a negative impact on their mental health.

Lastly, alcohol, tobacco, and consumption of other drugs were inversely correlated with eHLQ dimension one, “using technology to process health information”, dimension three, “ability to actively engage with digital services”, and dimension six, “access to digital services”; the first two in the overall and the last one in the younger group.

The literature found contradicts the results of this study; however, it only analyzed the relationship between healthy behaviors and digital health literacy in students belonging to faculties of health sciences [40,57] or university students in general [15] who did conclude that literacy in digital health is an important factor in health behaviors. However, the fact that the results are from such specific samples means that they are not as generalizable as those of this study; in addition, the results found in the literature could be questioned since they were only obtained with university students, not taking into account one of the most important variables in the influence of literacy in digital health are age and educational level. This conclusion is reinforced by the systematic review by Neter and Brainin [26], who observed that the literature analyzing this association was scarce and with inconsistent results.

This study possesses certain limitations. Given its cross-sectional nature, the ability to establish causality is restricted. Furthermore, as the research was conducted solely at a singular center catering to a demographic with medium-low socioeconomic status, the findings might predominantly apply to healthcare settings with similar attributes. The fact that only individuals who had the capability to visit the medical center were recruited in this study might mean that those who are not able to do so (and who potentially have lower health and eHealth literacy) were not included in this study. Conversely, this study employed a meticulous sampling approach, bolstering the internal validity of the outcomes. Additionally, this study’s inclusive approach to criteria for inclusion and exclusion enhances its external validity.

New lines of research are proposed that focus their efforts on analyzing the relationship between the social context and health behaviors and beliefs as determinants for a healthy lifestyle. The correlation and impact of having a diagnosed chronic disease on adhering to a healthy lifestyle compared to those who do not have one is also of paramount importance, in addition to those that focus on analyzing the impact of the barriers encountered by the population when trying to follow a healthy lifestyle. The need to carry out experimental studies that explore the causal relationship of health literacy and digital health on a healthy lifestyle is identified, as well as to create health education programs that focus on improving levels of health literacy and digital health in order to create a society without health inequalities. Therefore, research that analyzes healthcare settings and settings and how to ensure that they promote an adequate level of literacy and lifestyle is also timely.

## 5. Conclusions

This study contributes to our understanding of the relationship between health and eHealth literacy and a healthy lifestyle. Health literacy was mildly related to a healthy lifestyle with a higher impact on the younger population, with dimension three, “actively managing my health”, being the one with a bigger impact. eHealth literacy was mildly correlated with a healthy lifestyle, with dimension five, “motivated to engage with digital services”, having higher importance in the older population. These results imply that other factors might have a bigger correlation in having a healthy lifestyle. Being male or having been born in a third country had a negative repercussion on an individual’s lifestyle, whilst being married or retired/pensioner was a protective factor.

## Figures and Tables

**Table 1 healthcare-11-02980-t001:** Correlation between HLQ and “PA100” total score and HLQ and “PA100” dimension scores: overall and by age groups.

Variable		1. Feeling Understood and Supported by Healthcare Providers	2. Having Sufficient Information to Manage My Own Health	3. Actively Managing My Health	4. Social Support for Health	5. Appraisal of Health Information	6. Ability to Actively Engage with Healthcare Providers	7. Navigating the Healthcare System	8. Ability to Find Good Health Information	9. Understands Health Information Enough to Know What to Do
PA100	Overall	0.125 (*p* = 0.110)	0.275 **(*p* = 0.000)**	0.437 **(*p* = 0.000)**	0.190 **(*p* = 0.014)**	0.128 (*p* = 0.102)	0.233 **(*p* = 0.003)**	0.142 (*p* = 0.068)	0.072 (*p* = 0.355)	0.123 (*p* = 0.115)
	<65 years	0.143 (*p* = 0.203)	0.361 **(*p* = 0.001**)	0.516 **(*p* = 0.000)**	0.300 **(*p* = 0.007)**	0.377 **(*p* = 0.001)**	0.213 (*p* = 0.057)	0.206 (*p* = 0.065)	0.195 (*p* = 0.082)	0.229 **(*p* = 0.039)**
	≥65 years	0.101 (*p* = 0.357)	0.176 (*p* = 0.108)	0.304 **(*p* = 0.005)**	0.022 (*p* = 0.842)	0.080 (*p* = 0.466)	0.295 **(*p* = 0.006)**	0.152 (*p* = 0.165)	0.167 (*p* = 0.127)	0.182 (*p* = 0.095)
1. Healthy diet	Overall	0.017 (*p* = 0.829)	0.069 (*p* = 0.375)	0.175 **(*p* = 0.024)**	0.064 (*p* = 0.413)	0.250 **(*p* = 0.001)**	0.029 (*p* = 0.713)	−0.004 (*p* = 0.955)	0.136 (*p* = 0.080)	0.122 (*p* = 0.118)
	<65 years	0.177 (*p* = 0.113)	0.292 **(*p* = 0.008)**	0.357 **(*p* = 0.001)**	0.222 **(*p* = 0.047)**	0.449 **(*p* = 0.000)**	0.188 (*p* = 0.093)	0.142 (*p* = 0.205)	0.240 **(*p* = 0.031)**	0.224 **(*p* = 0.044)**
	≥65 years	−0.134 (*p* = 0.222)	−0.152 (*p* = 0.164)	−0.021 (*p* = 0.848)	−0.126 (*p* = 0.250)	0.156 (*p* = 0.153)	−0.092 (*p* = 0.400)	−0.111 (*p* = 0.310)	0.122 (*p* = 0.265)	0.095 (*p* = 0.386)
2. Physical activity	Overall	0.123 (*p* = 0.114)	0.219 **(*p* = 0.005)**	0.406 **(*p* = 0.000)**	0.168 **(*p* = 0.030)**	0.163 **(*p* = 0.036)**	0.190 **(*p* = 0.014)**	0.130 (*p* = 0.094)	0.032 (*p* = 0.682)	0.189 **(*p* = 0.015)**
	<65 years	0.115 (*p* = 0.305)	0.259 **(*p* = 0.019)**	0.444 **(*p* = 0.000)**	0.216 (*p* = 0.052)	0.320 **(*p* = 0.004)**	0.158 (*p* = 0.160)	0.163 (*p* = 0.146)	0.120 (*p* = 0.287)	0.278 **(*p* = 0.012)**
	≥65 years	0.132 (*p* = 0.229)	0.180 (*p* = 0.100)	0.347 **(*p* = 0.001)**	0.098 (*p* = 0.371)	0.097 (*p* = 0.379)	0.228 **(*p* = 0.036)**	0.136 (*p* = 0.214)	0.055 (*p* = 0.616)	0.174 (*p* = 0.112)
3. Alcohol, tobacco, and consumption of other drugs	Overall	0.009 (*p* = 0.910)	0.167 **(*p* = 0.032)**	0.172 **(*p* = 0.027)**	0.044 (*p* = 0.578)	−0.146 (*p* = 0.061)	0.027 (*p* = 0.727)	−0.001 (*p* = 0.990)	−0.017 (*p* = 0.833)	−0.086 (*p* = 0.269)
	<65 years	−0.031 (*p* = 0.781)	0.182 (*p* = 0.104)	0.164 (*p* = 0.143)	0.091 (*p* = 0.421)	0.037 (*p* = 0.742)	−0.003 (*p* = 0.981)	0.013 (*p* = 0.910)	0.124 (*p* = 0.272)	−0.026 (*p* = 0.816)
	≥65 years	0.064 (*p* = 0.558)	0.144 (*p* = 0.188)	0.117 (*p* = 0.285)	−0.058 (*p* = 0.599)	−0.145 (*p* = 0.184)	0.068 (*p* = 0.537)	0.030 (*p* = 0.783)	−0.006 (*p* = 0.954)	−0.031 (*p* = 0.782)
4. Emotional well-being	Overall	0.134 (*p* = 0.084)	0.164 **(*p* = 0.035)**	0.252 **(*p* = 0.001)**	0.251 **(*p* = 0.001)**	0.115 (*p* = 0.140)	0.318 **(*p* = 0.000)**	0.230 **(*p* = 0.003)**	0.042 (*p* = 0.590)	0.235 **(*p* = 0.002)**
	<65 years	0.258 **(*p* = 0.020)**	0.229 **(*p* = 0.040)**	0.297 **(*p* = 0.007)**	0.293 **(*p* = 0.008)**	0.288 **(*p* = 0.009)**	0.296 **(*p* = 0.007)**	0.268 **(*p* = 0.015)**	0.108 (*p* = 0.336)	0.286 **(*p* = 0.010)**
	≥65 years	0.007 (*p* = 0.950)	0.109 (*p* = 0.321)	0.182 (*p* = 0.096)	0.191 (*p* = 0.080)	0.054 (*p* = 0.625)	0.348 **(*p* = 0.001)**	0.216 (*p* = 0.047)	0.060 (*p* = 0.583)	0.270 **(*p* = 0.012)**
5. Safety and unintentional injuries	Overall	0.041 (*p* = 0.597)	−0.017 (*p* = 0.825)	−0.020 (*p* = 0.798)	−0.076 (*p* = 0.330)	−0.084 (*p* = 0.281)	−0.042 (*p* = 0.594)	−0.037 (*p* = 0.638)	−0.049 (*p* = 0.534)	−0.192 **(*p* = 0.013)**
	<65 years	0.083 (*p* = 0.460)	−0.051 (*p* = 0.649)	−0.048 (*p* = 0.669)	−0.076 (*p* = 0.498)	−0.120 (*p* = 0.284)	0.029 (*p* = 0.801)	0.103 (*p* = 0.360)	0.010 (*p* = 0.932)	−0.160 (*p* = 0.154)
	≥65 years	0.011 0.922	0.017 (*p* = 0.875)	−0.008 (*p* = 0.945)	−0.098 (*p* = 0.374)	0.031 (*p* = 0.778)	−0.109 (*p* = 0.323)	−0.151 (*p* = 0.167)	−0.027 (*p* = 0.803)	−0.164 (*p* = 0.134)

Health Literacy Questionnaire (HLQ). Correlation coefficients were measured as follows: very low (0 < r ≤ 0.19), low (0.2 < r ≤ 0.39), moderate (0.4 < r ≤ 0.59), high (0.6 < r ≤ 0.79), and very high (0.8 < r ≤ 1.0). Results in bold have a significant correlation (*p* < 0.05).

**Table 2 healthcare-11-02980-t002:** Correlation between eHLQ and “PA100” total score and eHLQ and “PA100” dimension scores: overall and by age groups.

Variable		1. Using Technology to Process Health Information	2. Understanding of Health Concepts and Language	3. Ability to Actively Engage with Digital Services	4. Feel Safe and in Control	5. Motivated to Engage with Digital Services	6. Access to Digital Services That Work	7. Digital Services That Suit Individual Needs
PA100	Overall	−0.032 (*p* = 0.678)	0.198 **(*p* = 0.011)**	−0.122 (*p* = 0.118)	0.102 (*p* = 0.191)	0.086 (*p* = 0.272)	0.081 (*p* = 0.300)	−0.038 (*p* = 0.631)
	<65 years	0.111 (*p* = 0.326)	0.309 **(*p* = 0.005)**	0.010 (*p* = 0.930)	0.159 (*p* = 0.155)	0.035 (*p* = 0.758)	0.102 (*p* = 0.365)	−0.023 (*p* = 0.842)
	≥65 years	0.191 (*p* = 0.081)	0.218 **(*p* = 0.045)**	0.123 (*p* = 0.263)	0.119 (*p* = 0.278)	0.287 **(*p* = 0.008)**	0.143 (*p* = 0.193)	0.198 (*p* = 0.070)
1. Healthy diet	Overall	0.186 **(*p* = 0.017)**	0.174 **(*p* = 0.025)**	0.046 (*p* = 0.560)	−0.007 (*p* = 0.925)	0.122 (*p* = 0.116)	0.038 (*p* = 0.628)	0.045 (*p* = 0.565)
	<65 years	0.334 **(*p* = 0.002)**	0.411 **(*p* = 0.000)**	0.113 (*p* = 0.316)	0.067 (*p* = 0.551)	0.209 (*p* = 0.061)	0.152 (*p* = 0.176)	0.099 (*p* = 0.380)
	≥65 years	0.257 **(*p* = 0.018)**	0.014 (*p* = 0.900)	0.107 (*p* = 0.330)	−0.087 (*p* = 0.429)	0.073 (*p* = 0.506)	−0.075 (*p* = 0.495)	0.028 (*p* = 0.802)
2. Physical activity	Overall	0.043 (*p* = 0.583)	0.202 **(*p* = 0.009)**	−0.041 (*p* = 0.602)	0.099 (*p* = 0.205)	0.145 (*p* = 0.062)	**0.167** **(*p* = 0.031)**	0.050 (*p* = 0.521)
	<65 years	0.235 **(*p* = 0.035)**	0.232 **(*p* = 0.039)**	0.056 (*p* = 0.620)	0.116 (*p* = 0.302)	0.206 (*p* = 0.065)	0.298 **(*p* = 0.007)**	0.104 (*p* = 0.356)
	≥65 years	0.075 (*p* = 0.495)	0.230 **(*p* = 0.034)**	0.043 (*p* = 0.693)	0.113 (*p* = 0.302)	0.180 (*p* = 0.098)	0.093 (*p* = 0.399)	0.119 (*p* = 0.280)
3. Alcohol, tobacco, and consumption of other drugs	Overall	−0.207 **(*p* = 0.008)**	−0.008 (*p* = 0.922)	−0.190 **(*p* = 0.014)**	0.032 (*p* = 0.680)	−0.089 (*p* = 0.255)	−0.098 (*p* = 0.211)	−0.139 (*p* = 0.073)
	<65 years	−0.092 (*p* = 0.413)	0.080 (*p* = 0.479)	−0.019 (*p* = 0.865)	0.078 (*p* = 0.490)	−0.170 (*p* = 0.128)	−0.225 **(*p* = 0.043)**	−0.133 (*p* = 0.236)
	≥65 years	−0.064 (*p* = 0.561)	0.002 (*p* = 0.987)	−0.025 (*p* = 0.823)	0.056 (*p* = 0.613)	0.123 (*p* = 0.263)	0.140 (*p* = 0.200)	0.073 (*p* = 0.505)
4. Emotional well-being	Overall	0.012 (*p* = 0.879)	0.190 **(*p* = 0.014)**	0.020 (*p* = 0.796)	−0.013 (*p* = 0.872)	0.050 (*p* = 0.523)	0.123 (*p* = 0.114)	0.062 (*p* = 0.425)
	<65 years	0.082 (*p* = 0.466)	0.220 **(*p* = 0.049)**	0.153 (*p* = 0.173)	0.005 (*p* = 0.966)	0.164 (*p* = 0.144)	0.316 **(*p* = 0.004)**	0.283 **(*p* = 0.010)**
	≥65 years	0.147 (*p* = 0.180)	0.220 **(*p* = 0.043)**	0.128 (*p* = 0.242)	−0.016 (*p* = 0.888)	−0.009 (*p* = 0.932)	−0.013 (*p* = 0.904)	−0.012 (*p* = 0.911)
5. Safety and unintentional injuries	Overall	−0.077 (*p* = 0.327)	−0.118 (*p* = 0.130)	−0.100 (*p* = 0.201)	−0.072 (*p* = 0.358)	−0.048 (*p* = 0.536)	−0.004 (*p* = 0.956)	−0.085 (*p* = 0.279)
	<65 years	−0.129 (*p* = 0.251)	−0.088 (*p* = 0.436)	−0.210 (*p* = 0.060)	0.012 (*p* = 0.914)	−0.059 (*p* = 0.603)	0.085 (*p* = 0.450)	−0.172 (*p* = 0.126)
	≥65 years	0.118 (*p* = 0.282)	−0.111 (*p* = 0.312)	0.166 (*p* = 0.129)	−0.141 (*p* = 0.198)	0.008 (*p* = 0.945)	−0.048 (*p* = 0.661)	0.128 (*p* = 0.244)

eHealth Literacy Questionnaire (eHLQ). Correlation coefficients were measured as follows: very low (0 < r ≤ 0.19), low (0.2 < r ≤ 0.39), moderate (0.4 < r ≤ 0.59), high (0.6 < r ≤ 0.79), and very high (0.8 < r ≤ 1.0). Results in bold have a significant correlation (*p* < 0.05).

**Table 3 healthcare-11-02980-t003:** Results of multiple regression analysis: predictors of a healthy lifestyle.

Predictors	Beta (95% CI)	*p*-Value
Global model R 0.543/R^2^ 0.295/adjusted R^2^ 0.273/F13.380		
Constant	42.615 (31.231, 53.999)	0.000
HLQ D3: actively managing my health	9.493 (5.692, 13.293)	0.000
Retired/pensioner	6.612 (2.803, 10,421)	0.001
Married	4.794 (0.855, 8733)	0.017
Country of birth: third country	−10.315 (−16.886, −3.744)	0.002
Sex: male	−6.854 (−10.830, −2.877)	0.001
<65 years model R 0.625/R^2^ 0.391/adjusted R^2^ 0.194/F9.616		
Constant	32.739 (13.309, 52.170)	0.001
HLQ D3: actively managing my health	15.362 (8.938, 21.786)	0.000
HLQ D2: having sufficient information to manage my health	7.504 (1.671, 13.336)	0.012
Separated	−12.611 (−21.318, −3.904)	0.005
eHLQ D6: access to digital services	−10.478 (−16.864, −4.091)	0.002
Sex: male	−5.765 (−11.380, −0.151)	0.044
≥65 years model R 0.348/R^2^ 0.121/adjusted R^2^ 0.099/F5.632		
Constant	66.670 (58.787, 74.552)	0.000
eHLQ D5: motivated to engage with digital services	5.233 (1.771, 8.695)	0.004
Sex: male	−4.956 (−9.841, −0.071)	0.047

Healthy lifestyle measured with the “PA100” questionnaire. Health Literacy Questionnaire (HLQ); eHealth Literacy Questionnaire (eHLQ); dimension (D).

## Data Availability

The data presented in this study are available upon request from the corresponding author.

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
