# Peer review of "Correlation between Health and eHealth Literacy and a Healthy Lifestyle: A Cross-Sectional Study of Spanish Primary Healthcare Patients"

_healthcare, 2023, doi:10.3390/healthcare11222980_

Round 1
Reviewer 1 Report (Previous Reviewer 2)
Comments and Suggestions for Authors
I think the content of the manuscript has been improved compared to the previous submission.
There are some points that need improvement, especially regarding statistical analysis.
This manuscript will require the supervision of a statistician.
1. About modeling multiple regression analysis. What is the author's purpose in performing multiple regression analysis? If the purpose is to compare the magnitude of influence on PA100, it would be sufficient to compare the magnitude of the correlation coefficient. Authors should clearly state the purpose of the multiple regression analysis in the methods section.
2. Multiple regression analysis is performed to test the multiple regression model. What is the multiple regression model that the author is going to consider? for example,
Global model: HLQD3+Retired/pensioner+Married+Country of birth: Third country+Sex: Male
I have a few more questions here.
1) Why is D3 the only independent variable here in HLQ? Why are other variables such as D2, D4, D6, and eHLQ not modeled as independent variables? The reasons should be made clear. However, the fact that the correlation coefficient is the highest among the HLQs is not evidence. I think It is best to make all correlated variables independent variables and then compare the magnitude of their influence. Even if the beta is not significant, it is necessary to interpret it carefully.
2) Isn't "Retired/pensioner" a category of the categorical variable "occupation"? The categorical variable "occupation" has five categories: employee, self-employed, unemployed, retired or pensioner, engaged in unpaid domestic work or student. When analyzing a categorical variable as an independent variable in multiple regression analysis, prepare dummy variables with the number of categories minus 1 and use it as an independent variable. This variable should create four dummy variables, but why aren't the other three dummy variables included in this model? The same goes for the "Separated" variable in the "<65" model.
3) Is “Third country” another country in L177? Or are you referring to another variable? Not only this variable, but all variables should be indicated using the same name as the variable name explained in the method and the variable name in the results.
4. What is the number in parentheses after beta in Table 3? Is it 95% CI?
Author Response
Please see the attachment

Reviewer 2 Report (New Reviewer)
Comments and Suggestions for Authors
Thank you very much for the opportunity to review the paper. It is a work that touches on a very interesting topic. Both the substantive and methodological preparation of the authors and the presentation of the results and the conclusions drawn from their analysis are very interesting. The work was prepared at a very high substantive, methodological, and analytical level.
I suggest to the authors that in the limitations of the paper, they add information regarding the small group size, which is also crucial for reading the results, especially in the oldest group of respondents over 65 years old.
The authors have prepared an excellent version of the paper for review, which is rare and testifies to the high scientific culture of the article's authors.
All the more, I recommend the paper for further publication process.
Author Response
Please see the attachment

Reviewer 3 Report (New Reviewer)
Comments and Suggestions for Authors
Dear authors, your research is very interesting.
I only have a few observations:
1. Correct link: 38 line
2. Check whether the following statement is correct: The sample was split into two age groups (81 <65 years and 85 ≥65 years old individuals) using the P50 (median) as a cut-off point.
3. Please revise the reference list according to the journal's requirements.
Good luck!
Author Response
Please see the attachment

Reviewer 4 Report (New Reviewer)
Comments and Suggestions for Authors
My thought is that even when people understand health information they may not act on it due to habits and preferences. Perhaps as you found, they only act on it if they have a chronic disease that implores them to do so.
Did you consider this in your research? If not, why not. I think that would confound your findings. Please respond.
Author Response
Please see the attachment

Reviewer 5 Report (New Reviewer)
Comments and Suggestions for Authors
Thank you for the opportunity to review this article.
The introduction provides a solid foundation for the study by clearly defining key concepts and establishing the relevance of the topic. However, it could benefit from clearer organization and more specific reference to the paucity of evidence on the relationship between health, e-health literacy and healthy lifestyle.
In the methodology, the authors provide a clear description of the study design, data collection procedures, and outcome measures.
Although the conclusion provides important information about the relationship between health, e-health and a healthy lifestyle, it could be improved by giving a direct response to the objective of the study, highlighting the main results and their relevance for practice and should present here suggestions for future investigations. The authors make suggestions, however, they present them in the discussion, and it is more appropriate to do so in the conclusion.
Congratulations to the authors.
Author Response
Please see the attachment

This manuscript is a resubmission of an earlier submission. The following is a list of the peer review reports and author responses from that submission.
Round 1
Reviewer 1 Report
Comments and Suggestions for Authors
Introduction was nicely written.
Including different Health Literacy scales adds value to the comparison
Some issues regarding the methodologies exist but may not be able to be corrected at this point since the data were collected; but they should be adequately described and also noted in the limitations section. Specifically,
· Participant’s average age is much older than the average age of the population. There could be a potential skew of the results regarding eHealth as older participants might have a lower eHealth score. Either redefine the target population as older adults or add this issue in Limitations.
· The distribution of <65 and >65 years old participants might present some potential bias as there are major age differences which can affect the health literacy of respective participants. Re-categorize the sample into more subgroups for analyses.
· Recruiting participants at a clinic only samples people that have the capability to visit a medical facility, thus contributing to potentially a higher health literacy score.
· How can a participant with an “illiterate” educational level complete the questionnaire in the study?
Results:
o Provide socio-demographic statistics to allow the readers to better understand the participant sample and help put the results in context.
· Tables 1 and 2 can be combined
o A graph showing distribution of the most significant results would help to better conceptualize the results
· The definition of “healthy lifestyle” and subsequent behaviors can vary based on different cultural understandings among the participants. Analyzing eHLQ and “PA100” total scores by sociodemographic variables might prove to be more insightful than by the two age group categories.
Comments on the Quality of English LanguageThe writing and English presentation are fine.
Reviewer 2 Report
Comments and Suggestions for Authors
This study examined the relationship between health literacy and healthy lifestyle for people using primary care.
However, there are some issues.
1. The introduction should explain the theoretical relationship between e-health literacy and health literacy. The theme of this research seems to be e-health literacy. However, both are mentioned in the results. Please explain the significance of daring to present both, and what results can be considered as hypotheses when setting objectives.
Do the authors expect the same result for both, or do they expect different results?
2. It is necessary to explain the reason for using HLQ and eHLQ in this study. First, there are many different health literacy scales. You are presenting HLS-EU and HLQ as commonly used scales, but please clearly show the rationale for them. And please explain why HLQ and not HLS-EU. Please do the same for eHLQ.
3. Since the subjects of this study were receiving primary care, what was the distribution of their diseases and their history of illness?
In addition, it is necessary to show the frequency distribution of age and socio-demographic variables.
4. Control variables in multiple regression analysis should be unified. Theoretically, it should not be possible for the model to have different control variables for reasons other than the multicollinear relationship.
5. The discussion is redundant. The first paragraph continues from L269 to L380. You need to do paragraph writing. The discussion should be developed under the intention of hypothesis testing, rather than developing the argument as an afterthought.
Round 2
Reviewer 2 Report
Comments and Suggestions for Authors
Unfortunately, the author seems to have done very little to correct my previous comment.
1.“The introduction should explain the theoretical relationship between e-health literacy and health literacy. The theme of this research seems to be e-health literacy. However, both are mentioned in the results. Please explain the significance of daring to present both, and what results can be considered as hypotheses when setting objectives.”
I still don't understand why the authors use both e-health literacy and health literacy in this study. If the hypothesis presented by the authors, use only eHLQ and do not need to use HLQ. Arguments are inconsistent.
2.I made a comment ”It is necessary to explain the reason for using HLQ and eHLQ in this study. First, there are many different health literacy scales. You are presenting HLS-EU and HLQ as commonly used scales, but please clearly show the rationale for them. And please explain why HLQ and not HLS-EU. Please do the same for eHLQ ”, but the author only says “The following justifies the selection of the questionnaires utilized to measure health and eHealth literacy in comparison to others that measure similar aspects.”
But I still don't understand why they don't use HLS-EU.
3. I made a comment “Since the subjects of this study were receiving primary care, what was the distribution of their diseases and their history of illness?”
but the author only says “These data were not collected during the recruitment process.”
I consider the content of my comment to be a fatal problem in this research, but the author does not particularly think so. The author does not intend to specifically modify or review.
4.I made a comment “Control variables in multiple regression analysis should be unified. Theoretically, it should not be possible for the model to have different control variables for reasons other than the multicollinear relationship.”
However, the authors did not make any corrections and did not appear to revisit their analysis. Given the multiple regression analysis and multivariate adjustment, researchers should carefully consider what their control variables are. In some cases, we perform stratified regression analysis. Any analytical model should be adjusted with the planned control variables.